# Reproducibility study of "Data-Driven Methods for Balancing Fairness and Efficiency in Ride-Pooling"

## Reproducibility Summary

**Scope of reproducibility**

Our work attempts to verify two methods to mitigate forms of inequality in ride-pooling platforms proposed in the paper *Data-Driven Methods for Balancing Fairness and Efficiency in Ride-Pooling* [1]: (1) integrating fairness constraints into the objective functions and (2) redistributing income of drivers. We extend this paper by testing for robustness to a change in the neighbourhood selection process by using actual Manhattan neighbourhoods and we use corresponding demographic data to examine differences in service based on ethnicity.

**Methodology**

The authors of the paper provide preprocessed data and code implemented in TensorFlow, which we transform into PyTorch. Experiments in this reproducibility study can be divided into 3 parts: (1-2) we reproduce the results regarding objective functions and income redistribution using data and settings provided in the paper and code; (3) we apply this approach to the same data grouped into Manhattan neighbourhoods. Further, we examine discrepancies between service rates of different ethnicities using neighbourhood-specific demographic data as a proxy for this protected information.

**Results**

The results in the original paper regarding different objective functions were reproduced within a margin of error. Also, income redistribution is able to reduce wage inequality, albeit to a lesser degree. The objective functions appear to be sensitive to the neighbourhood selection mechanism. While the results of the rider-fairness objective functions are maintained, performance of the driver-fairness objective functions declines. There appear to be only small differences in service rates between ethnicities, while rider-side fairness seems to mitigate inequalities the most. However, this is only achieved by worsening the service for well-served neighbourhoods instead of improving it for underserved ones.

**What was easy**

The simulation logic as well as the training and testing procedures in the provided code were straightforward to execute.

**What was difficult**

To be able to run the authors' code we needed to make several changes to it. Moreover, specific parts of the original research were not explicitly mentioned in the paper. Another point of difficulty was the absence of preprocessing code which was not detailed properly and could not be fully reproduced. The reproducibility of the paper relied on the provided code, communication with the authors as well as previous works.

**Communication with original authors**

We contacted the authors about the preprocessed data that was not hosted online due to licensing issues. They supplied it as well as responded very quickly and provided clarifications on the parameters and their values in the code.

Submitted to ML Reproducibility Challenge 2021. Do not distribute.

# 1 Introduction

Ride-pooling, where drivers can service multiple requests from riders simultaneously, is becoming increasingly popular [2]. Since resources are shared, ride-pooling has the potential to reduce the aggregate VKT ("vehicle kilometres travelled") and with that reduce petroleum usage and carbon dioxide emissions [3]. To efficiently perform the matching of riders and drivers, machine learning algorithms are used [4], which optimise for income maximisation. However, with respect to ride pooling, previous works have observed a gender wage gap [5] as well as majority Asian and Hispanic neighbourhoods being associated with less service compared to white neighbourhoods [6]. Therefore, alternative fairness notions could also be useful.

Shah et al. [7] introduces an algorithm to solve the ride-pooling matching problem, which maximises the number of rider requests serviced based on a Markov decision process (MDP) in combination with deep learning. The authors of the paper *Data-Driven Methods for Balancing Fairness and Efficiency in Ride-Pooling* [1] extend this work to compare multiple objective functions, defined on different fairness metrics. Next to that, they investigate the use of income redistribution. In this reproducibility study, we attempt to verify their results and extend their experiments.

# 2 Scope of reproducibility

The main contribution of the paper is introducing and evaluating measures to deal with the fairness issues arising in ride-pooling. In our reproducibility study, we first focus on reimplementing their code (implemented in TensorFlow [8]) in PyTorch [9] and compare the results we achieve to their findings. The main claims made in the original paper are:

- The authors claim that they extend the MDP-based framework (introduced in [7]) by incorporating different definitions of fairness to perform non-myopic optimisation. By incorporating fairness measures into the objective function, driver and rider inequality can be reduced while maintaining or even improving profitability.
- The state-of-the-art objective function [7] can outperform the fairness objective functions in certain settings in terms of rider-fairness and increase the average income of drivers at the cost of a higher variance.
- Income redistribution can be used to reduce wage inequality while avoiding the free-rider problem and guaranteeing a minimum wage for drivers.

The mathematical proof guaranteeing the minimum wage is not verified in our study. In addition to testing for reproducibility, we examine the robustness of the approach to changes in the neighbourhood selection method using actual tabulation areas. Using demographic data, we investigate whether the fairness objective functions are fair to all ethnicities. To investigate these aspects of the paper, we followed these steps:

1. We inspect the provided codebase and identify, analyse and solve any barriers to running the code.
2. Next, we transform the code to the PyTorch framework, matching the functionality as well as possible.
3. With the PyTorch version we attempt to reproduce the results using the dataset preprocessed by the authors. To investigate potential differences, we use different seeds to examine the effect of randomness.
4. To test the method's robustness we utilise the authors' approach on actual neighbourhoods in Manhattan and, using the neighbourhood demographic compositions (since individual protected data is confidential), we explore whether the introduced objective functions mitigate potential inequalities between ethnic groups.

# 3 Theoretical background

The paper we are reproducing extends the method proposed in [7]. The latter presents Neural Approximate Dynamic Programming (NeurADP), which uses offline-online learning and approximates dynamic programming to match drivers and riders non-myopically. The following subsections explain NeurADP and the two extensions proposed in [1], fairness-based objective functions and income redistribution.

## 3.1 NeurADP: Neural Approximate Dynamic Programming

NeurADP uses neural network-based value function approximation and updates it using the Bellman equation [10]. To break temporal dependencies between samples, mini-batch experience replay is used [11].

The neural network is used to rank feasible actions for each agent. To receive the optimal choices, an integer linear program (ILP) is solved considering the top 150 feasible actions. To update the neural network, the authors use a target network and Double Q-learning [12]. The value function over individual vehicles is learned offline. When the approach is running online, the model computes the driver-rider assignment that maximises the value function computed in the offline phase. Further details regarding the neural network inputs and its architecture are in Appendix A.

## 3.2 Fairness-based objective functions

Prior work used profitability metrics as objective functions. The authors introduce two new objective functions to improve both driver-side and rider-side fairness [1] and compare them using different evaluation strategies.

**Profitability objectives** There are two profitability measures used: the number of riders serviced ($o_1$) and the total income ($o_2$).

$$o_1(R,W) = \sum_{i=1}^{n} |p_i| + |s_i|, \qquad o_2(R,W) = \sum_{i=1}^{n} \underbrace{\sum_{u \in p_i \cup s_i} E_{g,e} + \delta}_{\pi_i} \tag{1}$$

The total number of rides serviced by driver $i$ consists of the number of ongoing requests $|p_i|$ and completed requests $|s_i|$. The total income is calculated by adding the incomes $\pi_i$ of the individual drivers $i$. The income for any request $u$ is the sum of the variable cost $E_{g,e}$ (depending on the start and end locations $g$ and $e$) and the fixed part of ride-pooling pricing, represented by the constant $\delta$.

**Fairness objectives** The authors define two fairness metrics for rider-side ($o_3$) and driver-side ($o_4$) fairness.

$$o_3(R,W) = -\lambda \text{Var}\left(\frac{h_j}{k_j}\right) + \sum_{i=1}^{n} \pi_i \qquad o_4(R,W) = -\lambda \text{Var}(\pi_i) + \sum_{i=1}^{n} \pi_i \tag{2}$$

The former is quantified by the variance of the success rates which is computed by the ratio between serviced and total requests $\left(\frac{h_j}{k_j}\right)$ originating in neighbourhood $j$. Each crossing is mapped to one of $H$ neighbourhoods. $o_4$ is based on the spread of incomes $\pi_i$. Both objective functions incorporate the total income $o_2$ into the equation, $\lambda$ controls the importance of the variance term.

**Evaluation strategy** To measure the effect of different objective functions, the authors introduce two fairness metrics. They evaluate rider-fairness by comparing the overall and minimum success rates across neighbourhoods. By contrast, they utilise the income distributions to assess driver-fairness.

## 3.3 Income redistribution

The authors also introduce an income redistribution scheme to mitigate income fluctuation and inequality in driver wages. To help estimate the true contribution of each driver, Shapley values [13] are used. In this ride-pooling setting, a Shapley value can be intuitively interpreted as the average profit lost when a specific driver does not contribute.

To reduce the difference between a driver's pre-redistribution income $\pi_i$, and Shapley value $v_i$, the authors use a risk parameter, $0 \leq r \leq 1$, which designates what fraction of a driver's income is kept. The model collects $\sum_{i=1}^{n}(1-r)\pi_i$ from all drivers and redistributes it proportional to the difference between their value and earnings, which is $max(0, v_i - r\pi_i)$. The driver's income after redistribution, $q_i$, is

$$q_i = rv_i + \frac{max(0, v_i - r\pi_i)}{\sum_{j=1}^{n} max(0, v_j - r\pi_j)} \sum_{j=1}^{n} (1-r)v_j \tag{3}$$

**Evaluation strategy** To measure the correlation between the Shapley value and income after redistribution, the gain metric $g_i$ is defined as the ratio of change in $q_i$ to $v_i$ when $v_i$ is doubled. The gain $g$ is calculated as the average over $g_i$. To test the effect of income redistribution, the authors determine gain and the standard deviation of the ratio of $q_i$ to $v_i$ for varying values of $r$. The most desirable outcome is that the driver's redistribution value is as close as possible to their Shapley value, i.e. $std = 0$ and that if they double their contribution, they double their earnings after redistribution, i.e. $g = 1$.

# 4 Methodology

In this section, the approaches used in our reproducibility study are outlined.

## 4.1 Datasets

The following shows the original dataset and the demographic data to the Manhattan neighbourhoods.

### 4.1.1 NYC yellow taxi data Manhattan

Similar to [1], we use the dataset 'Yellow taxi trip records' from New York City [14] for training and evaluation. The original dataset contains pick-up and drop-off coordinates for taxi passengers. We follow the assumption of the original paper that the spatial and temporal distribution of rider requests between ride-pooling and taxi rides are similar. The preprocessing done in [1] consists of the following steps. First, the dataset of New York City is filtered to only comprise trips starting and ending in Manhattan. Next, the coordinates are discretised into $|L|$ locations, which are identified by taking the street network of the city from openstreetmap [15] using osmnx with 'drive' as network type. We take the largest strongly connected component of the network discarding nodes that do not have outgoing edges.

The resulting network has 4373 locations (street intersections) and 9540 edges. The pick-up time is converted to batches of requests corresponding to the minutes. Furthermore, the locations are grouped into 10 neighbourhoods using K-means clustering [16]. The dataset contains on average 322714 requests in a day (on weekdays) and 19820 requests during the peak hour. The preproccessed dataset was not publicly available, although mentioned otherwise in the paper. The authors confirmed that this was due to licensing issues and provided us with the preprocessed data. The model is trained using the data from March 26th - 28th 2016. The fairness objective functions are tested on the data from April 4th.

### 4.1.2 Demographics by Neighborhood Tabulation Area

The dataset "Demographics by Neighborhood Tabulation Area" for New York City [17] allows us to investigate whether the ride demand of racial or ethnic minorities is indeed satisfied in the same way. It contains demographic data for each neighborhood tabulation area (NTA) in New York City. A NTA is an area for which census data is gathered. The demographic data relevant to this report are the race/ethnicity percentages per neighbourhood, namely Hispanic/Latino, White, Black/African-American, Asian, Other. Instead of running K-means clustering to obtain the neighbourhoods, we take the neighbourhoods corresponding to these NTA areas in Manhattan. This results in 29 instead of 10 neighbourhoods for Manhattan. To be able to determine which nodes in the graph are situated in which NTA, we made use of the "2010 Neighborhood Tabulation Areas" dataset [18] which contains coordinates specifying an approximation of the polygon shape of each neighbourhood.

## 4.2 Code

Our implementation is based on the code of the paper which is publicly available at GitHub [1]. The repository was updated after we started reproducing the paper, but we refer to the commit specified above unless stated otherwise. The published code is not functioning and does not include the preprocessing steps. However, the main framework for testing and training is provided and hyperparameters can be configured using setting files. We re-implemented the model in the PyTorch framework [9], ensuring that the default behaviour of TensorFlow which was implicitly used in the authors' implementation is replicated. This includes weight initialisation and hyperparameters of the optimiser. To transfer the masking mechanism used to pad the sequences, we employed PyTorch's packed sequence implementation. Since the new framework does not support backwards LSTM, we used a bidirectional LSTM and ignored the forward pass to achieve the same functionality. In accordance with the original code, we used the CPLEX optimiser [19] to solve the ILP. To support the number of drivers and, therefore, bigger linear systems, the academic or commercial version is necessary. There were some rare situations in which the ILP failed to satisfy the constraints (one or two agents were not assigned any actions) which led to an error. This was fixed by assigning the "take no action" action to those agents. In addition, we implemented the preprocessing steps on the original dataset found at [14], as this code

---

[1]https://github.com/naveenr414/ijcai-rideshare/tree/78d81d0f417ad4fd54ea2e967010bb221fc4e177

was not available. For this, we perform the same steps indicated in Section 4.1.1, but we simplified the estimation of the travel times as this was not clear from the paper. Our code is available at GitHub. [2]

## 4.3 Hyperparameters

Focusing on reproducing the original paper [1], we tried to stay close to the original paper's approach and did not perform hyperparameter optimisation. Hyperparameter values missing in the paper (e.g. minimum number of experiences and samples) were retrieved from the authors' code. Additionally, there were inconsistencies, when some hyperparameters had different values in different parts of code (e.g. embedding dimension). In this case, we reached out to the authors for clarification. More details on hyperparameters are in Appendix C.

## 4.4 Computational requirements

To increase the available computational resources, we used multiple computers with different hardware (see Table 5 in the Appendix). In general, the training time is dominated by the simulation of the environment and solving the ILP. The training of the neural network plays only a minor role. Hence, GPUs are not crucial for training, the training time is mostly determined by the single-core performance of the CPU. A run consisting of training on three days and testing on one typically takes about 2.5 to 3 hours. In total, running all experiments took 202 hours.

## 4.5 Experimental setup

**Experiment 1** To reproduce the results regarding claims 1 and 2, different settings are needed, presented in Table 1. All combinations of these settings are used. The requests and income objective functions do not have lambda values. Furthermore, the embeddings are trained (further details are in Appendix A.1). We use the same training/testing split as in the paper (described in Section 4.1), and evaluate the results based on overall and minimum success rates as well as income distribution.

| Setting | Value |
|---|---|
| Number of drivers | 50 , 200 |
| Objective function | Driver, rider, requests, income |
| Lambda | Driver: 0, 1/6 , 2/6 , 3/6 , 4/6 , 5/6 , 6/6
Rider : $10^8, 10^9, 10^{10}$ |
| Training days | 3 |
| Testing days | 1 |

Table 1: Settings used for the experiments

To test if the differences between our findings and the original results are caused by randomness, we rerun the experiments using different seeds. Due to limited resources, we rerun only a subset of setting combinations. Further details can be found in Appendix B.

**Experiment 2** In accordance with the original paper, the results of the first experiments are reused to evaluate the income redistribution for claim 3. The analysis is focused on the 200 drivers with the requests objective function using gain and standard deviation (see Section 3.3).

**Experiment 3** To test robustness we use the 29 predefined neighbourhoods and train the models using the configurations of experiment 1 for only 200 drivers. To incorporate the demographic data for the analysis presented in step 4 (see Section 2), we map the results per neighbourhood to the five different ethnicities, under the assumption that the distribution of ethnicities living in a neighbourhood corresponds to the distributions of riders' ethnicities. For each group, we calculate the mean across all neighbourhoods weighted by the percentage of this group living in that area. This results in five different values per objective function. The higher this value is, the more requests of the corresponding group are serviced. Since we are interested in the difference across groups, we subtract the average of this rate. Values above zero indicate a group that is serviced above average and, hence, could be interpreted as advantaged. In addition, we evaluate the overall, minimum and per neighbourhood success rates.

## 5 Results

In the following, we will present the results of the three different experiments.

---

[2]https://github.com/reproducibilityaccount/reproducing-ridesharing

### 5.1 Reproducibility result 1 - Fairness objective functions

Looking at our findings in Figure 1, we conclude that for 50 drivers the results for the rider-fairness metric can be reproduced, the success rates for the different objective functions match. Using the driver-fairness objective function improves both the success rate and the rider equality.

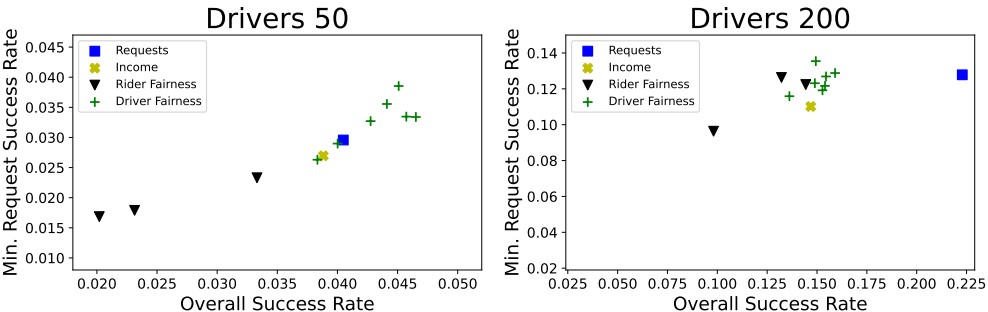

Figure 1: Comparison of objective functions for different number of drivers, $\lambda$ not included, reflecting the original paper

For 200 drivers, there are minor discrepancies between our results and the original. They can, however, be explained by stochasticity introduced by different seeds. However, for rider-fairness with $\lambda = 10^{10}$, the difference can not be explained by randomness. The requests objective function often results in more profit and better rider equality.

For each objective function, the payment distribution for 200 drivers is shown in Figure 2. The variance of the distributions are similar in magnitude, the means however are slightly shifted. Looking at the differences between the results for different

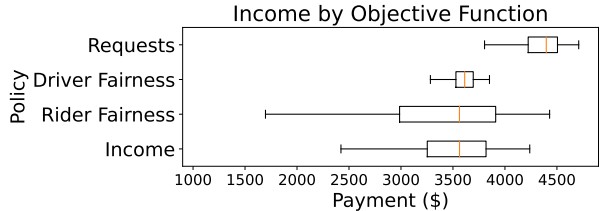

Figure 2: Comparison of income distributions ($\lambda = \frac{4}{6}$ for driver-side fairness and $\lambda = 10^9$ for rider-side fairness)

seeds, this could be explained by randomness. The driver-fairness objective function is able to reduce the variance in income between drivers, but the profitability is also decreased. Appendix E shows the results presented in the original paper, the results of the different seeded runs are visualised in Figure 10.

### 5.2 Reproducibility result 2 - Income redistribution

The authors' findings regarding the effect that varying the risk parameter $r$ has on the gain and the standard deviation of the ratio $\frac{q_i}{v_i}$ were not reproducible on the basis of the information in the paper alone, nor were they immediately reproducible from the code itself. Upon further communication with the authors, they updated their code. There was also a typo in the formula given in Equation 3 (Equation 12 in [1]). The correct equation is:

$$q_i = r\pi_i + \frac{max(0, v_i - r\pi_i)}{\sum_{j=1}^{n} max(0, v_j - r\pi_j)} \sum_{j=1}^{n} (1 - r)\pi_j, \quad (4)$$

where it can be seen that the use of Shapley values in the first term and last factor have been replaced by the amounts before redistribution. With these corrections in place, our experiments yielded the results seen in Figure 3. For values of $0.4 \leq r \leq 0.6$ the gain is non-zero whilst maintaining a spread close to zero for the redistribution income to Shapley value ratio. In the original paper, this condition held for values of $0.5 \leq r \leq 0.9$. Furthermore, the magnitude of the gain is far smaller at the point at which the spread begins to increase. This indicates that when $r = 0.6$, drivers only receive a $40\%$ increase in their wages whilst still earning close to their true contribution. This is in contrast with the original, where, for $r = 0.9$, drivers receive an $80\%$ increase in their wages while minimising the free-rider problem. This leads us to conclude that the results of this redistribution scheme were not reproducible in this setting. The original results are shown in Figure 8 in the Appendix.

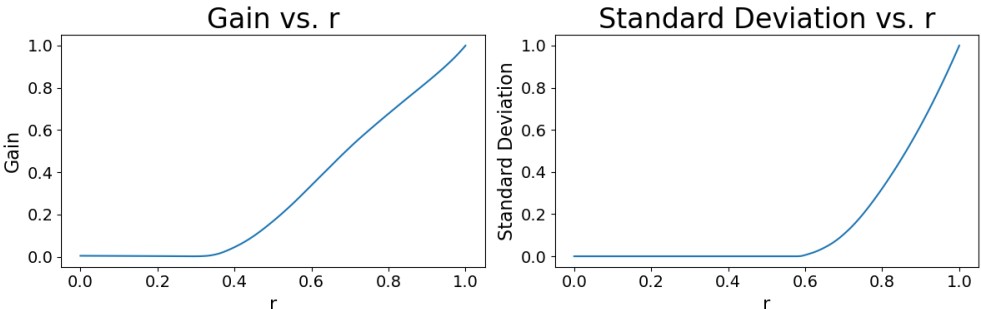

Figure 3: Comparison of the gain metric and the standard deviation of the income to value ratio for different values of risk parameter r

## 5.3 Results for Manhattan neighbourhoods and incorporating demographic data

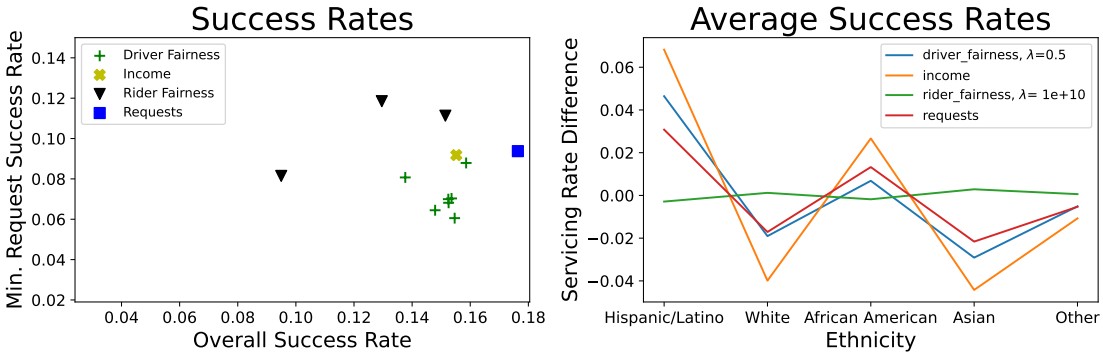

Figure 4: Analysis of results incorporating demographic data

We retrained the model (see Section 4.5). Comparing the resulting Figure 4 to previous findings in Figure 1, we observe that by changing the neighbourhoods the performance of the driver-fairness objective functions deteriorates the most. The rider-fairness objective functions share some similarities between the two experiments but the latter now performs best in terms of fairness across neighbourhoods (minimum request success rate).

The right plot in Figure 4 shows that there are small differences in the percentage of requests serviced per ethnicity. The rider-fairness objective function for $\lambda = 10^{10}$ seems to be best at mitigating inequality. However, as seen in the left plot, rider-fairness results in low success rates. This might indicate that the objective function merely lowers success rates for otherwise well-serviced neighbourhoods rather than improving under-serviced ones.

To confirm this, we visualised the success rate per neighbourhood and objective function (see Figure 5). It can be seen that rider-fairness indeed exhibits notably reduced variance but also a lower mean when compared to the other objective functions which tend to have an upward skew. This shows that rather than benefiting under-serviced neighbourhoods, applying rider-fairness only lessens the success rate of well-served ones.

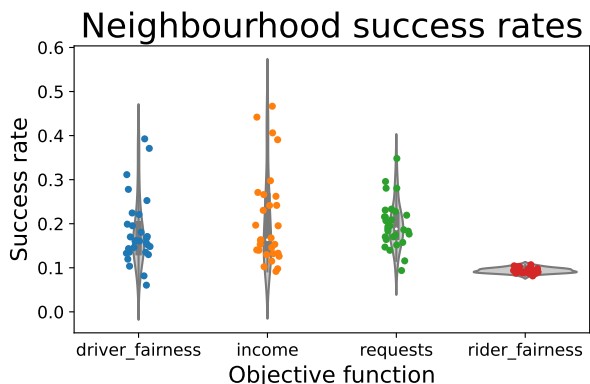

Figure 5: Success rate per neighbourhood ($\lambda = 0.5$ for driver-fairness, $\lambda = 10^{10}$ for rider-fairness)

# 6  Discussion

Combining the results from the reproducibility experiments (experiments 1 and 2 in Section 4.5), we find that the first claim mentioned in Section 2 is supported by our results for 50 drivers. Furthermore, our results substantiate the second claim. The 'requests' objective function can improve the rider-fairness for 200 drivers. Additionally, it results in the highest average income per driver but exhibits a higher variance than the driver-fairness objective function. These observations are in accordance with the ones of the original paper.

For the 200 drivers setting, specific results were more sensitive to sources of stochasticity than for 50 drivers. After inspecting the code, we found that the minimum number of experiences needed to start the training of the neural network is never exceeded for the 50 drivers setup. In the 200 drivers configuration, it is reached and hence the neural network is trained. Since the weights of the model are randomly initialised, it might converge to a different local minimum which yields a different value function. This could explain the variance in the corresponding results. For 50 drivers, in contrast, no learning is involved. Hence, the result goes through a randomly initialised model. Weights are typically initialised to preserve the mean and variance of the input, which should be unaffected by the specific seed used. This could explain the strong similarity between our results and the original results for the 50 drivers setup.

Differences found in reproducing the income redistribution scheme may also be accounted for by the above. However, while our results are not exactly the same, the third claim still holds, although to a considerably lesser degree than in the original paper.

When employing the actual Manhattan neighbourhoods, the relative standing of the various objective functions was different compared to the ones determined by K-Means. This indicates that the proposed method is sensitive to the neighbourhood selection mechanism. Looking at the demographic data, it can be seen that all objective functions exhibit small differences between ethnicities. These, however, could be attributed to stochasticity.

In any case, rider-fairness results in the least variance across ethnicities at the price of mean success rate. However, this result does not imply that rider-fairness achieves this low variance by better servicing neighbourhoods with a lower percentage of accepted requests, but rather by servicing better-served neighbourhoods less well. Importantly, the ethnicity-based analyses are built on the assumption that the distribution of the ethnicities of residents and riders in a neighbourhood is similar. However, ride pooling might be used by other people like commuters or tourists. Furthermore, there could be differences between the ethnic populations regarding the percentage of ride-sharing users.

## 6.1  What was easy

Part of the code, namely the simulation logic, did not need any modifications. This logic is responsible for telling drivers of possible rides to accept as well as executing the drivers' choices and keeping the simulation consistent with respect to the existing constraints. The training and testing procedure was also straightforward to execute.

## 6.2  What was difficult

The codebase was not originally executable and required modifications. In addition to that, several aspects of the original research were not explicitly mentioned in the paper. Although, in the end, we were able to reproduce most results, this would not have been possible without consulting either the code, the authors or the paper about NeurADP [7]. Another challenge was the absence of preprocessing code which together with the lack of a detailed description in the paper (specifically for travel time estimates) made its implementation difficult. With the limited time resources we had, we did not succeed in testing if our preprocessing implementation affected the results.

## 6.3  Communication with original authors

The authors were very helpful, kind and responded very quickly, often within the same day. This was a very important factor in the production of this reproducibility report as the preprocessed data could not be hosted online due to licensing issues. Furthermore, they also provided useful clarifications with respect to the parameters used in the code and discrepancies between different parameter values in different places. The authors also updated the codebase following our discussions.

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

# Appendix

## A    Neural network details

The inputs to the neural network model are the current location of the vehicle, the information about the remaining delay, and locations for the current requests that have been accepted. First, authors order them according to their trajectory and feed them as inputs to an LSTM [20] after an embedding layer. The embeddings for the locations are calculated separately and are the byproduct of a two-layer neural network that attempts to estimate the travel times between two locations (see Appendix A.1).

Additional inputs to the neural network are the information about the current decision epoch, the number of vehicles in the vicinity of the vehicle of interest and the total number of requests that arrived in the epoch. This information is used to stabilise learning because the value of being in a given state is dependent on the competition it faces from other drivers when it is in that state. These inputs are concatenated with the output of the LSTM from the previous paragraph and, after 2 dense layers, used to predict the value. An overview of the details of the neural network can be seen in Table 2.

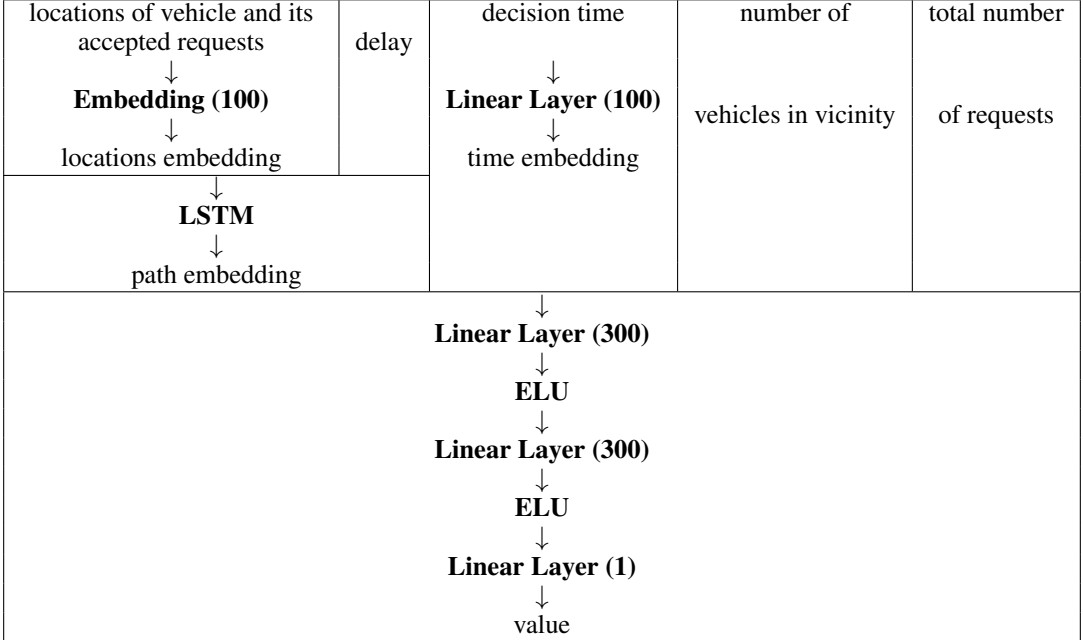

Table 2: Overview of the value approximation neural network. The model layers (with output dimensions in brackets) are presented in **bold**.

### A.1 Embeddings training

In accordance with paper [7], the embedding model, shown in Table 3, was trained for 1000 epochs with batch size 1024 and Adam optimiser with default settings. The training also utilises early stopping with patience 15.

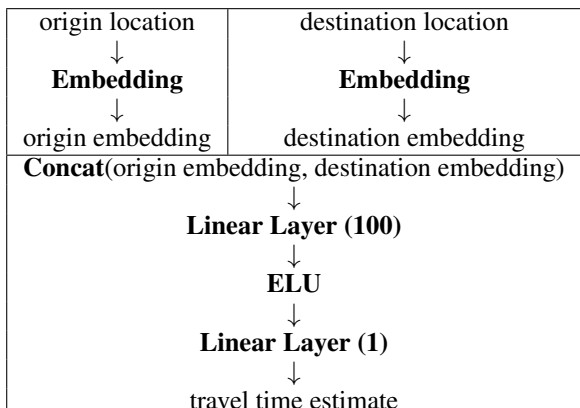

Table 3: Embedding Model. The model layers (with output dimensions in brackets) are presented in **bold**.

## B Seeds

The settings selected for seeded runs, have to meet several conditions. First of all, we wanted to rerun at least one setting for all four objective functions. Next to that, for the rider-fairness, we rerun all lambda values because this objective function differed the most between our results and the original paper's. For the driver-fairness, we only chose a lambda value of 4/6, since all lambda values yield similar results and only this one is used to examine both, driver- and rider-side fairness metrics.

By default, the seed 874 is used. If further seeds are used for experiments, the following four are utilised: 688701, 490013, 423376, 191758.

## C Hyperparameters

| Hyperparameter names | Values |
|:---:|:---:|
| number of locations: $|L|$ | 4461 |
| number of neighbourhoods: $H$ | 10 |
| max. capacity of driver: $m$ | 4 |
| ride-pooling pricing: $\delta$ | 5 |
| pick up delay | 300 |
| drop off delay | 600 |
| min. replay buffer size | $5 * 10^5$ / (number of riders) |
| number of samples | 3 |
| gamma: $\gamma$ | 0.9 |

Table 4: Hyperparameter values.

# D Hardware configurations

| Name | CPU | GPU | RAM |
|---|---|---|---|
| Setup 1 | i5-8600k | GTX1080 | 16 GB |
| Setup 2 | i7-1165G7 | - | 32 GB |
| Setup 3 | Apple-M1 | - | 16 GB |
| LISA cluster[a] | Intel Xeon Silver 4110 | GTX1080 Ti | 32 GB |

Table 5: Hardware configurations used.

[a]One Nvidia GTX1080Ti GPU with 3 CPUs provided by SURFsara's LISA cluster. For more info see: `https://userinfo.surfsara.nl/systems/lisa/description`

# E Results of the original paper

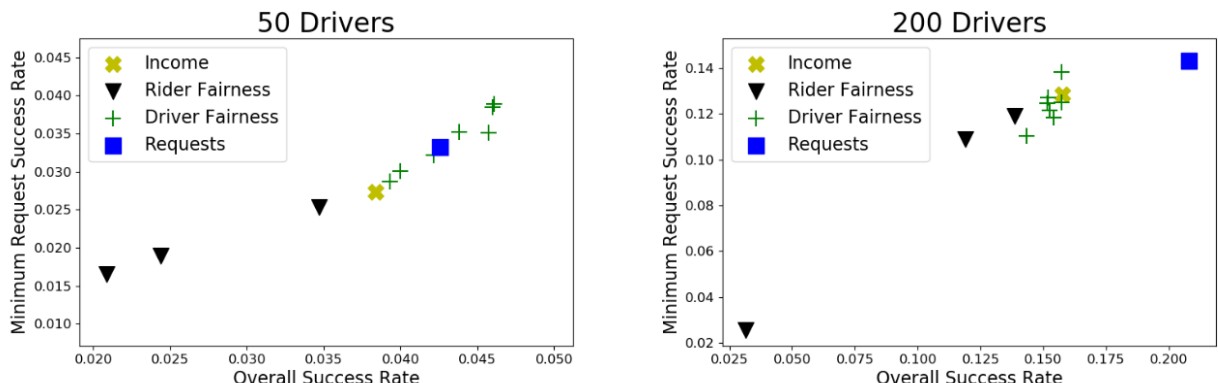

Figure 6: Figure of the original paper [1] comparing objective functions for different number of drivers.

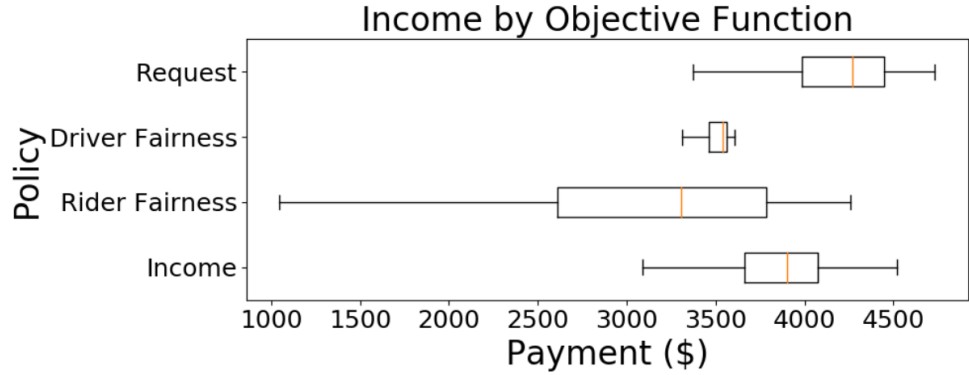

Figure 7: Figure of the original paper [1] comparing the distribution of incomes for different objective functions ($\lambda = \frac{4}{6}$ for driver-side fairness and $\lambda = 10^9$ for rider-side fairness.)

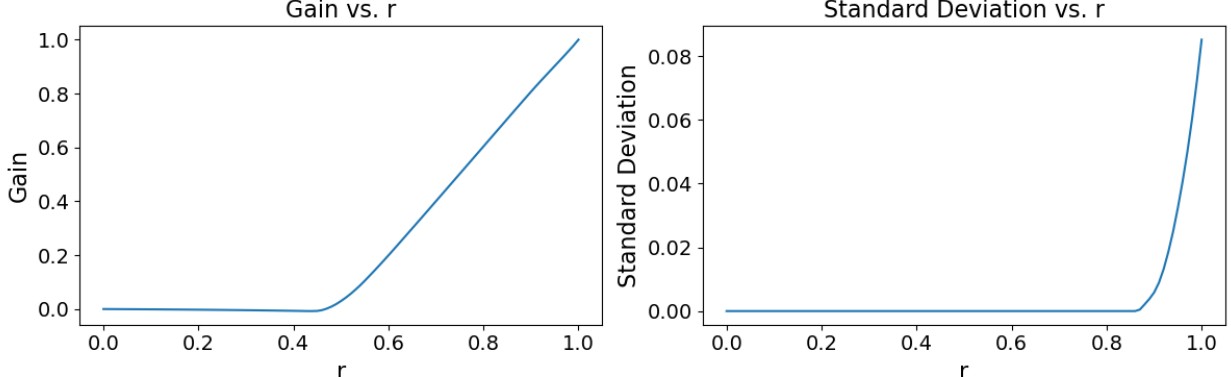

Figure 8: Figure of the original paper [1] comparing the gain metric to the standard deviation of the redistributed income to Shapley value ratio for different values of r.

367 **F    Results for different seeds**

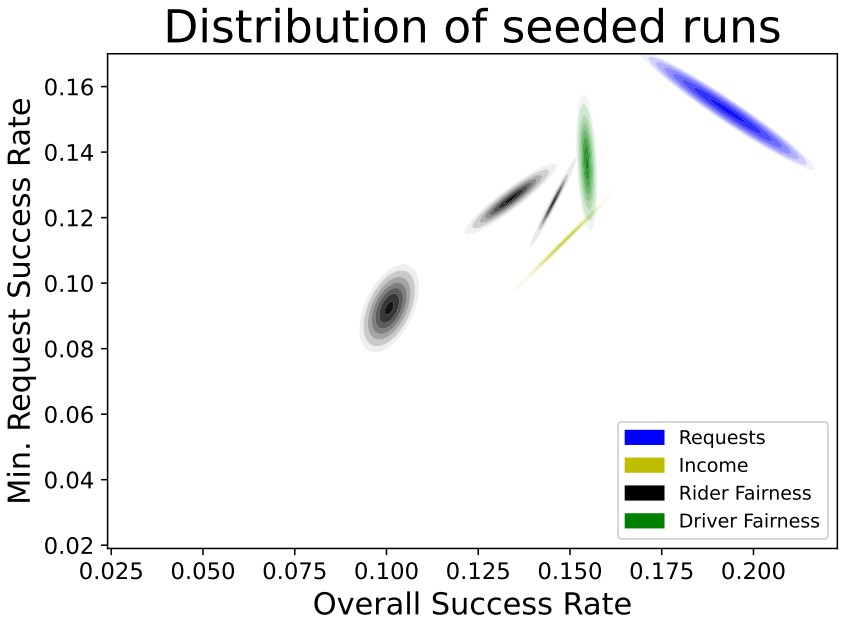

Figure 9: Comparison of objective functions for 200 drivers with five different seeds. Each configuration is modelled as a bivariate Gaussian distribution. The $\lambda$ values for the rider-fairness are (from left to right): $10^{10}, 10^9, 10^8$, for the driver-fairness: $\lambda = \frac{4}{6}$.

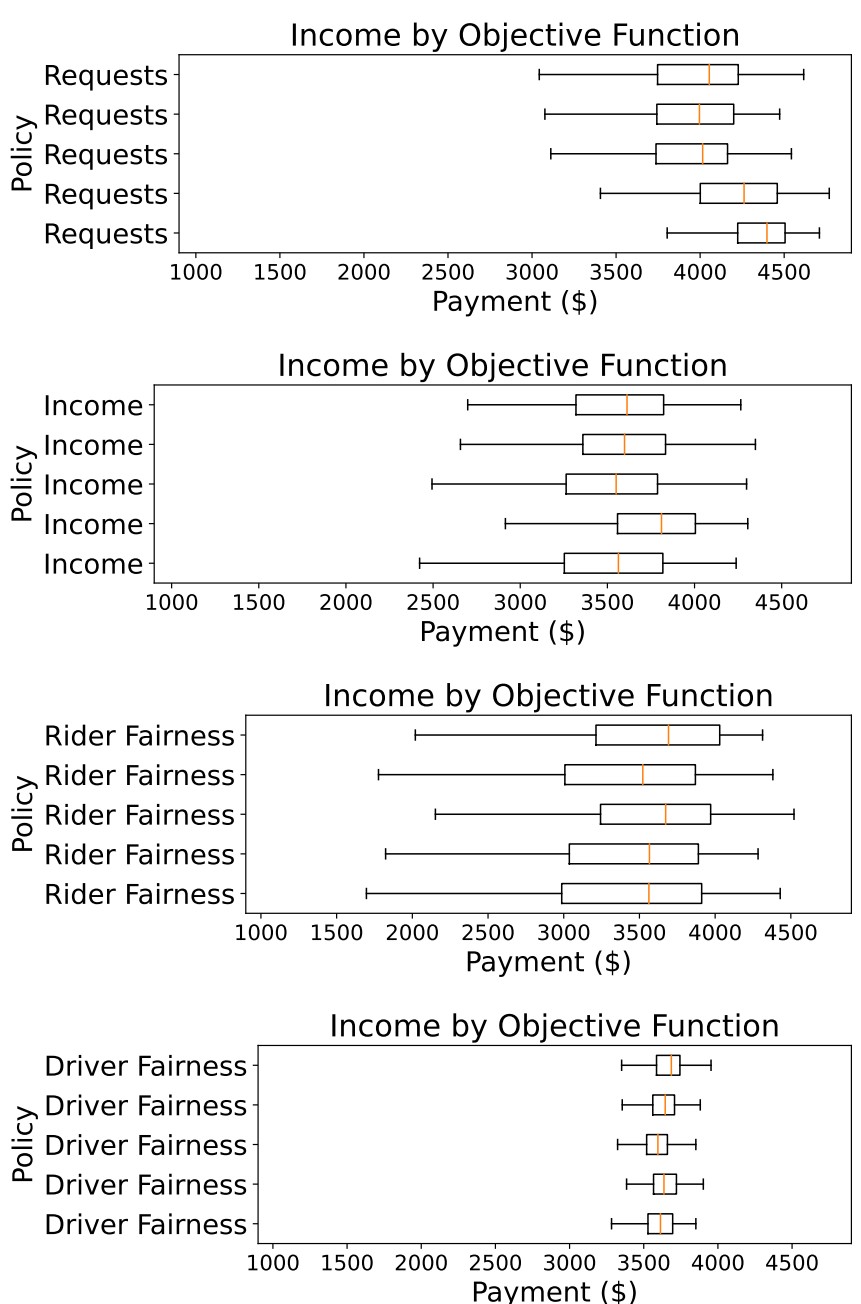

Figure 10: Comparing the distribution of incomes for different objective functions with five different seeds ($\lambda = \frac{4}{6}$ for driver-side fairness and $\lambda = 10^9$ for rider-side fairness.)

