# OpenReview forum: "Reproducibility study of "Data-Driven Methods for Balancing Fairness and Efficiency in Ride-Pooling""
_ML_Reproducibility_Challenge/2021/Fall — RC2021_

### Official Review · Reviewer_dJp2 · 2022-03-07
**Good Replication.**

**Rating:** 8
**Confidence:** 4

**Review:**

Summary: This is a very strong reproduction. The authors do a nice job of (a) providing an overview of the original work, (b) detailing their reproduction effort and (c) summarizing their findings. I found the decomposition fo what was /wasn't reproducible with hypotheses around why to be particularly valuable. The authors do a nice job of not treating the reproduction effort as a binary yes/no problem, and treat it as a nuanced problem describing both the latent weaknesses and strengths of the reproduced paper along the way.

Code: Authors implement in pytorch and attempt to follow the original authors' and hyperparameter search.

Discussion on results: Authors do a commendable job of discussing the implications of their findings at each step.

Recommendations for reproducibility: No explicit recommendation for reproducibility.

Results beyond the paper : None, though I think this is reasonable in this case.

Overall organization and clarity: No clear grammatical issues. Good writing and organization.

---

### Official Review · Reviewer_oaP4 · 2022-03-28
**Useful reproducibility report, but can benefit from better writing and presentation**

**Rating:** 6
**Confidence:** 4

**Review:**

Overall, this is a useful reproducibility report, particularly with the usage of neighborhood data to ascertain the various fairness contributions made by the original paper. However, the report can significantly benefit from improved writing, especially by shedding motivation on the choice of extended experiments, providing insight on the implications of these findings, and overall increasing the amount of detail in the manuscript.

* **Reproducibility Summary:**

The report contains a reproducibility summary incorporating their main findings and the overall replicability of the original paper.
Scope of reproducibility:
The authors consider several empirical results from the original paper. This includes replicating main results within a margin of error and applying the techniques presented—objective functions that incorporate fairness and schemes for income redistribution—to the same setting, but with data grouped according to Manhattan neighborhoods. Finally, in their submission, the authors do not delve into veracity of the theoretical contributions made.


* **Code:**

Authors re-implement the code from scratch in Pytorch, which was originally open-sourced in Tensorflow. This certainly strengthens the reproducibility study and permits the authors to cover the nitty-gritty of the original algorithm.


* **Communication with original authors:**

It is positive to note the communication with the original authors that made it possible to fill the necessary gaps needed for replication, such as hyperparameter values and the pre-processing steps.


* **Hyperparameter Search:**

Hyperparameter values were obtained either from the original paper (when accessible), or from the open-sourced code base, or by communicating with authors when necessary. No additional hyperparameter tuning was carried out.


* **Ablation Study:**

No ablation study has been conducted.


* **Discussion on results:**

The authors provide a fair amount of detail in this section. Supplementary Figure 9 seems to be more valuable than Figure 1 for illustrating results. I would encourage the authors to replace both subplots in Figure 1 by their equivalents (in Suppl. Fig. 9 that incorporates multiple random seeds). Going further, consider adding standard deviation bars to plots such as Figure 4 (right).

Considering the very high value of $\lambda=10^9, 10^{10}$ used for rider-side fairness, can the second term in $o_3$ be ignored?

Has there been a discussion with the original authors about the discrepancy noted for income redistribution (Sec 5.2)?

Perhaps a bar plot will be more appropriate than a line plot in Figure 4 (right), as the categories are not continuous-valued. Please consider renaming “experiment 1, 2, 3” with more meaningful headers.

Instead of “right plot in Figure 4”, please use subplots to refer to relevant portions.


* **Recommendations for reproducibility:**

The authors find certain contributions of the paper reproducible within a reasonable margin of error, while finding contributions pertaining to income redistribution not holding to the claimed extent.

Line 126 -> was the original authors' pre-processed data used for reproducibility experiments or were the preprocessing steps listed in lines 123–125 implemented afresh for the purpose of this study?


* **Results beyond the paper:**

Grouping results by neighborhood data results in an interesting finding—driver fairness as an objective reduces variance across ethnic groups, but primarily seems to be do so by reducing success rates in existing well-serviced neighborhoods. This questions the utility of the objective function proposed by the original authors.

It would be interesting to see the difference between the neighborhoods found from the KMeans algorithm versus the actual Manhattan neighborhoods. This would certainly strengthen the existing report.


* **Overall organization and clarity:**

The paper is fairly well-organized, but there is significant scope to enhance readability.

Please avoid using citation numbers as nouns. Equation 2 can be made more readable. Use “\max” or “\text{max}” in Equation 3. Consider using booktabs to format tables. All figures could use more detailed captions—talk about the objective of the experiment, how to read the plot and the implications of these plots. Further, consider removing the existing plot titles as they do not convey any new information: use subcaptions instead. A few incorrect usages such as “by contrast” (line 94) can be corrected.

---

### Meta-Review · Area_Chair_LUPd · 2022-04-08

**Recommendation:** Accept
**Confidence:** 4

**Metareview:**

A good reproducibility study with a detailed analysis of the original paper. The author's effort in writing the code from scratch in PyTorch is commendable. It would be great to work on some presentation issues that reviewers suggested.

---

### Decision · Program_Chairs · 2022-04-09

**Decision:**

Accept

**Comment:**

Following the recommendation of reviewers and meta-reviewer, the paper is accepted for ML Reproducibility Challenge 2021, and will be published in the upcoming special edition of ReScience Journal.